# HC-Net: Memory-based Incremental Dual-Network System for Continual learning

## Abstract

Training a neural network for a classification task typically assumes that the data to train are given from the beginning. However, in the real world, additional data accumulate gradually and the model requires additional training without accessing the old training data. This usually leads to the *catastrophic forgetting* problem which is inevitable for the traditional training methodology of neural networks. In this paper, we propose a memory-based continual learning method that is able to learn additional tasks while retaining the performance of previously learned tasks. Composed of two complementary networks, the Hippocampus-Net (H-Net) and the Cortex-Net (C-Net), our model estimates the index of the corresponding task for an input sample and utilizes a particular portion of itself with the estimated index. The C-Net guarantees no degradation in the performance of the previously learned tasks and the H-Net shows high confidence in finding the origin of an input sample.

## 1 Introduction

The main difference between the human brain and the machine learning methodology is the ability to evolve. Using neurophysiological processing principles, human brains can achieve and organize knowledges throughout their lifespan. Having the neuroplasticity, human brains can transfer an activating region of a given function to a different location or control the creation and destruction of synapses according to its experiences. Usually, artificial neural network (ANN) models consist of finite numbers of filters. Also, parameters and operations in the ANN do not possess the ability corresponding to the memory system of human brains. This structural limit leads to the problem called *catastrophic forgetting*, i.e., the newly coming information diverts the model from previously learned knowledge. The field of researches trying to solve this problem is referred to as *continual* or *lifelong learning*.

Kirkpatrick et al. (2017) suggested a method applying a penalty on the quadratic distance between the old parameters and the new ones. Called elastic weight consolidation (EWC), this method can be applied to either supervised or reinforcement learning scenarios. Jung et al. (2018) proposed a method using feature regularization referred as less-forgetful learning (LF). When training a new dataset, the feature just before the fully connected layer is regularized to become similar to that of the previous model. A considerably small learning rate is used in order to prevent a drastic change of the model. Despite of such efforts, as more and more tasks are added, the model eventually collapses with poor performances in old datasets. Li & Hoiem (2017) proposed a method using a similar regularization method, referred as learning without forgetting (LwF). Continual learning is conducted to a single convolutional neural network (CNN), adding an additional classifier whenever a dataset is newly trained. Classifiers in the newly trained network are regularized to reproduce the outputs of the old network while the new classifier is trained in a classical way. LwF shows better performance than LF, but still suffers from the catastrophic forgetting with multiple tasks. These approaches using regularization basically focus on alleviating the deformation of the original network. Therefore, the more the new knowledge flows in the model, the more the performance on the original task degrades. To make a single network deal with a large number of datasets, instead of using all parameters of a single network for each task, PackNet suggested a model which allocates the datasets to specific weights of filters (Mallya & Lazebnik, 2018). As only the specialized weights for a particular task are involved in its classification process, PackNet shows a remarkable result in multiple tasks. However, the information about from which task the input originates and which

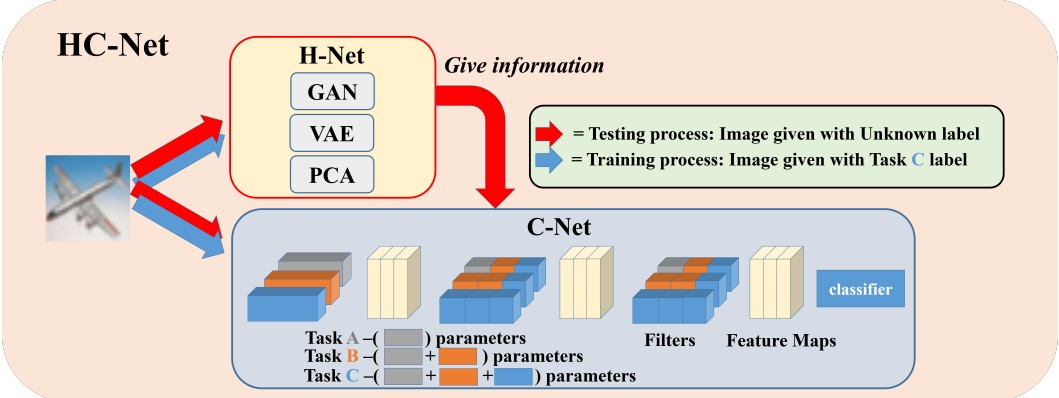

Figure 1: The overall architecture of our method. Blue arrows depicts the training process for the new task (Task C) of our method, where Task A and Task B have been already trained. H-Net is trained each time a dataset for a new task is provided and then the C-Net is trained in a standard way such as using the cross-entropy loss. We allow only the trainable filters (Blue parameters) to be updated. Red arrows shows the inference procedure of our method. H-Net finds out the task index $\mathcal{J}$ (Task C) from which the input came and let the C-Net to use the filters specialized for task $\mathcal{J}$ (Blue + Orange + Grey). Class labels are also switched to those of task $\mathcal{J}$.

group of weights should be used must be given in advance. Typically, images do not contain such prior knowledges and this makes the PackNet hard to apply in real world situation.

In this paper, inspired by the human brain system, we propose a network which efficiently increases its complexity without degrading the performance of previous tasks. Instead of saving all the memory in the cortex and trying to access it directly, the hippocampus finds the index of the memory scattered over the cortex where knowledges are well organized inside. Under the inspiration from the quote in (Buzsaki, 2006), "Think of the cortex as a huge *library* and the hippocampus as its *librarian*", we have built our network using two complementary sub-networks as shown in Figure 1. The *Cortex-Net* (C-Net) keeps the knowledge from several independent tasks. After training a network from the previous task, additional filters in the convolutional modules are attached and trained with the next task. To infer the newly learned data, the model utilizes the newly trained filters along with the previously learned filters, whereas the data from the old task is inferred only using the previously learned filters. In order to determine which combination of filters to use, we adopted a memory network which can distinguish the origin of a given input sample. Referred as the *Hippocampus-Net* (H-Net), it can recall the information not only which group of filters to use but also which group of class labels to use. This endows our method label-expandability, which enables our method to be practically applied in real world situation. We suggest three methods, each of which respectively using generative adversarial networks (GAN) (Goodfellow et al., 2014), variational auto-encoder (VAE) (Kingma & Welling, 2013), and principal component analysis (PCA) (Jolliffe, 2011) for H-Net and report the properties and performance of each method. Our H-Net can be combined with any other works such as LwF or PackNet. The combined H-Net and C-Net referred as HC-Net can prevent the *catastrophic forgetting* using a densely organized architecture and a highly reliable memory system. The contributions of this paper can be summarized as follows:

- Filters in the convolutional layers are allocated to each task and parameters from the previously learned tasks are shared among post tasks and used for the initialization of post tasks parameters, which makes the overall model compact and efficient.

- Attaching filters whenever a new dataset is trained allows the model to be expandable as far as the physical constraints, such as memory and processing time constraints, allow.

- To identify which portion of convolutional filters should be used to classify a data, a memory network (H-Net) is proposed with several methodologies.

- The overall method is applicable to multi-task continual learning whose number of class labels increases over tasks.

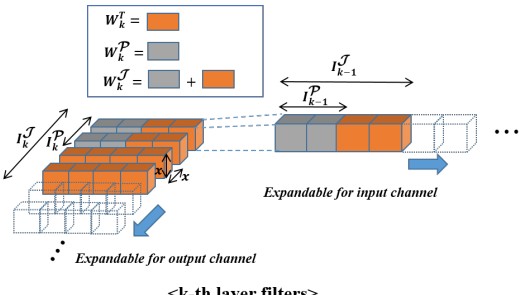

Figure 2: Trainable parameters $W_k^T$ (Orange) and frozen parameters $W_k^{\mathcal{P}}$ (Grey) in the C-Net training process. In the training process, the parameters (Orange), which are trainable parameters except frozen parameters (Grey), are updated. $x$ denotes the spatial dimension of filters.

## 2 METHOD

We propose a memory-based incremental dual network system for continual learning that satisfies three properties as stated below.

**Property 1.** The *Hippocampus-Net* (H-Net) offers the index about which part of the *Cortex-Net* (C-Net) should be used.

**Property 2.** The C-Net learns the knowledge on the new task under the presence of the previously learned knowledge.

**Property 3.** H-Net and C-Net are flexible to the expansion of the network and as many new tasks can be learned in one network as physical constraints allow.

In this respect, our proposed H-Net works as a hippocampus of a human brain in that it gives a summarized information to the C-Net. Also, the C-Net corresponds to the human cortex in terms of its ability to learn semantic knowledge.

### 2.1 CORTEX NETWORK (C-NET)

We propose an efficient way to allocate the capacity to a network according to the given tasks. C-Net is an expandable network where additional filters can be attached whenever a new task is given. It can be applied to many typical networks such as VGG and ResNet (Simonyan & Zisserman, 2014; He et al., 2016). The only difference with the original versions of those networks is that it uses different parts of filters for different tasks. As shown in (Li et al., 2016), typical CNN are capable of maintaining their performance even when the majority of their parameters are pruned. This implies that filters in a CNN contains a lot of unnecessary or redundant information. Also, sharing the filters among different tasks inevitably leads to the *catastrophic forgetting* of the network since the parameters trained by an old task should be replaced with those trained by a new task. For these reasons, we utilize a network divided into several parts and refer it as C-Net where each part takes charge of a particular task.

While training the C-Net with several tasks, each task uses a different part of the C-Net. For the convenience of referring which segments of the C-Net are to be activated, we introduce a filter index $I^{\mathcal{J}}$ for the $\mathcal{J}$-th task. The filter index $I^{\mathcal{J}}$ defines the range of filters to use for the specific task $\mathcal{J}$ in every layers. For example, if the filter index $I^1$ equals to ten, the first task uses filters from the first to the tenth filter.

Depicted in Figure 2, $I_k^{\mathcal{J}}$ denotes the filter index in the $k$-th layer corresponding to the task $\mathcal{J}$. This equals to the number of output channels in the sequent feature map. $I_k^{\mathcal{P}}$ denotes the filter index in the $k$-th layer for the previous task. As the same principle is applied to the precedent $(k-1)$-th layer, the number of input channels for the filters in the $k$-th layer will change likewise. That is, total of $I_k^{\mathcal{P}}$ filters have the channel length of $I_{k-1}^{\mathcal{P}}$ when the previous task is under training. As the new task is given, the length of the filters become $I_{k-1}^{\mathcal{J}}$ since the filters in the prior layer will deliver a feature

---

**Algorithm 1** C-Net Training

---

**Input:** $W^{\mathcal{J}}, I^{\mathcal{P}}, I^{\mathcal{J}}$
**Output:** $W^{\mathcal{J}*}$
  1: Freeze the $W^{\mathcal{P}} \subset W^{\mathcal{J}}$ using the information $I^{\mathcal{P}}$
  2: Initialize the $W^{\mathcal{T}}$ with $W^{\mathcal{P}}$ and a random noise
  3: $W^{\mathcal{J}*} \leftarrow \arg\min(\mathcal{L}_c)$
     {Update $W^T$ using backpropagation}

---

map whose length is $I_{k-1}^{\mathcal{J}}$. Therefore, both the number of input and output channels increases as the number of tasks increases until the physical constraints allow.

Weight parameters of $\mathcal{J}$-th task in filters of the $k$-th layer are referred as $W_k^{\mathcal{J}} \in R^{x \times x \times I_{k-1}^{\mathcal{J}} \times I_k^{\mathcal{J}}}$, where $x$ means the size of the kernel while $W_k^{\mathcal{P}} \in R^{x \times x \times I_{k-1}^{\mathcal{P}} \times I_k^{\mathcal{P}}}$ are the weight parameters of the previous task. When training the new task, $W_k^{\mathcal{J}}$ is used for inference but only the parameters $w \in W_k^{\mathcal{T}}$ ($W_k^{\mathcal{T}} = W_k^{\mathcal{J}} \setminus W_k^{\mathcal{P}}$) are updated while leaving $W_k^{\mathcal{P}}$ fixed. Note that biases in the filters can also be trained likewise as explained above. The fully connected layer located just before the linear classifier can be divided as well into the sections of $I^{\mathcal{J}}$ and $I^{\mathcal{P}}$.

To enhance the performance, slices of trained parameters $W^{\mathcal{P}}$ from the old task are duplicated to the newly available task parameters $W^T$ and Gaussian noise is added to them for the sake of good initialization. Then, the network is trained in a standard way, e.g., using a cross-entropy loss $\mathcal{L}_c$. Independent linear classifiers are used for every tasks. This scheme of C-Net can also be applied to structures using shortcut connections such as ResNet. The overall training process is shown in Algorithm 1.

## 2.2 HIPPOCAMPUS NETWORK (H-NET)

Even if the C-Net is well trained separately with several tasks, the task index $\mathcal{J}$ must be given at the time of inference. Especially when the class labels expand, this prior information $\mathcal{J}$ is necessary to estimate which group of class labels to use. In the testing step, existing methods needs the exact origination of the input data. However in the real world, a given data for classification normally does not offer any information about its origination. Therefore, this prior knowledge should be estimated by an independent network.

To solve this problem, we introduce the Hippocampus network (H-Net) using several different methods (GAN, VAE and PCA), which is able to estimate the task $\mathcal{J}$ of the given input data and inform this to the C-Net to specify which filters to use. In this paper, we explain the GAN method as a representative and other methods are explained in Appendix A.

### 2.2.1 GENERATIVE ADVERSARIAL NETWORKS (GAN)

This method is inspired by the on-line replay method used for continual learning (Shin et al., 2017). We train a task-specific generative model $G_{\mathcal{J}}$ using a GAN to generate pseudo-samples of task $\mathcal{J}$. The generator is trained using the adversarial loss as follows:

$$\min_{G_{\mathcal{J}}} \max_{D_{\mathcal{J}}} V(D_{\mathcal{J}}, G_{\mathcal{J}}) = \mathbb{E}_{x \sim P_{data}^{\mathcal{J}}}[\log(D_{\mathcal{J}}(x)] + \mathbb{E}_{z \sim p_z(z)}[\log(1 - D_{\mathcal{J}}(G_{\mathcal{J}}(z))]. \quad (1)$$

Here, $D_{\mathcal{J}}$ is a discriminator for the task $\mathcal{J}$ and $x$ is a sample from the task $\mathcal{J}$.

After generating samples of all tasks, a task-wise binary classifier $B_{\mathcal{J}}$ is trained to classify whether the given input is from the task $\mathcal{J}$ or not. Note that only the generated samples are involved for the training of each binary classifier $B_{\mathcal{J}}$. This means that positive samples are from the generator $G_{\mathcal{J}}$ for the task $\mathcal{J}$ and negative samples are from all the other task generators $G_{\mathcal{K}}$ ($\forall \mathcal{K} \neq \mathcal{J}$).

In the testing phase, each classifier produces the probability of how likely the given unknown input data is from the task $\mathcal{J}$. The classifier $B_i$ with the highest probability assumes that the input data is from the task $i$. H-Net can figure out the task $\mathcal{J}$ of input data with maximum probability across the

set of classifiers. H-Net gives this estimated task index $\mathcal{J}$ to the C-Net. The H-Net using GAN can be summarized as below:

$$
\begin{aligned}
\text{Initialization:} \quad & B = \{B_1, B_2, ..., B_N\} \\
\text{Training:} \quad & B_i(x) = \begin{cases} 1 & \text{if } x \text{ is from } G_i \\ 0 & \text{if } x \text{ is from } G_j \; \forall j \neq i \end{cases} \\
\text{Inference:} \quad & \mathcal{J} = \arg\max_i B_i(x)
\end{aligned}
\tag{2}
$$

## 3 EXPERIMENT

We evaluate our method on several image classification datasets. First, we verify the effectiveness of our method with MNIST (LeCun et al., 1998), SVHN (Netzer et al., 2011) and CIFAR-10 (Krizhevsky & Hinton, 2009) which are widely used to evaluate image classification performance. Then, we evaluate our method on two subsets of ImageNet (Russakovsky et al., 2015) which is a realistic image dataset. More details are summarized in Table 1.

We compare our method to various methods such as Elastic Weight Consolidation (EWC), Learning without forgetting (LwF), less-forgetful learning (LF) and PackNet as well as networks trained for a single target task which shows the performance without conducting continual learning. We use the GAN method for H-Net in all the experiments as a representative. To solely examine the influence of H-Net on our method (HC-Net), we have conducted experiments with the same experimental scenarios on the HC-Net experiments. In Table 2, each row shows the result of one among GAN, VAE and PCA. In Table 3 and Table 4, the result of 'HC-Net (without H-Net)' shows the performance of the C-Net only. That is, it assumes that the H-Net never fails and the task index $\mathcal{J}$ is always given correctly. Likewise, LwF and PackNet need prior knowledge of a given input to know which classifier to use. As there is no structure like H-Net in the original paper, The actual comparison must be done with 'C-Net'.

### 3.1 CONTINUAL LEARNING USING BASIC IMAGE CLASSIFICATION DATASETS

We have built three experimental scenarios to evaluate our method using basic image classification datasets. We compare the performance of learning a consecutive two task pair among MNIST, SVHN and CIFAR-10 dataset. After that, we conduct multiple-task continual learning with these datasets. In these experiments, we use the same learning hyperparameters and network architectures suggested in (Jung et al., 2018) used for 'Tiny image classification'. Training details are reported in Appendix B.

#### 3.1.1 MNIST → SVHN

After selecting a subset of SVHN to equalize the number of training data between MNIST and SVHN, images are resized to $28 \times 28$ as in (Jung et al., 2018). The results of Modified LwF, EWC and LF are referred from (Jung et al., 2018). Modified LwF is sligthly different from the original LwF in that the old output $\hat{Y}_o$ and new outputs $\hat{Y}_n$ is united into a single output. The 'single network' in Table 3 is a network of full capacity trained using a single dataset. The column 'Old' represents the test accuracy on the previously learnt task and 'New' is that of the newly trained task . The value $\gamma$ for EWC is determined as in (Kirkpatrick et al., 2017). A knowledge distillation loss with hyper-parameters of $T = 2$ and $T = 1$ is used for LwF, referred as temperature in (Hinton et al., 2015).

As MNIST and SVHN highly resemble each other, the accuracy of all methods are relatively high. LF demonstrate better average classification rates than Modified LwF and EWC. LwF can control the performance trade-off between the old and new task by changing the temperature parameter $T$. The bigger $T$ is, the lower the accuracy drops but it still outperforms those of the methods mentioned above. PackNet almost maintains the performance of the single network for the first task with less than 2% drop in accuracy. HC-Net outperforms all other methods in the table. Note that the prior knowledge to select which task is given in experiments of other methods. Therefore, our HC-Net without H-Net (i.e., C-Net), which shows the highest in performance, should be compared to other

| Datasets | #of Train data | #of Test data | #of Class |
|---|---|---|---|
| MNIST | 60,000 | 10,000 | 10 |
| SVHN | 73,275 | 26,032 | 10 |
| CIFAR-10 | 50,000 | 10,000 | 10 |
| ImageNet-A | 64,750 | 2,500 | 50 |
| ImageNet-B | 64,497 | 2,500 | 50 |

Table 1: Details of datasets.

| Datasets | Methods | Old(%) | New(%) |
|---|---|---|---|
| MNIST | GAN | 100 | 99.94 |
| ↓ | VAE | 100 | 99.85 |
| SVHN | PCA | 100 | 100 |
| SVHN | GAN | 97.00 | 99.90 |
| ↓ | VAE | 71.07 | 97.06 |
| CIFAR-10 | PCA | 93.28 | 92.5 |

| Datasets | Methods | MNIST(%) | SVHN(%) | CIFAR(%) |
|---|---|---|---|---|
| MNIST | GAN | 100 | 96.65 | 99.83 |
| → SVHN | VAE | 100 | 71.22 | 97.07 |
| → CIFAR-10 | PCA | 100 | 93.28 | 92.5 |

Table 2: Experimental results of H-Net on the basic datasets

| Datasets | Methods | Old(%) | New(%) | Avg. (%) |
|---|---|---|---|---|
| MNIST | single network (MNIST) | 99.49 | – | – |
| | single network (SVHN) | – | 92.82 | – |
| ↓ | Modified LwF | 94.78 | 83.77 | 89.28 |
| SVHN | EWC ($\gamma = 2.32 \times 10^4$) | 94.15 | 79.31 | 86.73 |
| | LF ($\lambda_e = 1.6 \times 10^{-3}$) | 97.37 | 83.79 | 90.58 |
| | LwF ($T = 1$) | 98.27 | 86.40 | 92.34 |
| | LwF ($T = 2$) | 97.33 | 86.97 | 92.15 |
| | PackNet | 99.45 | 91.49 | 95.47 |
| | HC-Net | 99.43 | 92.20 | 95.82 |
| | HC-Net (without H-Net) | 99.43 | 92.37 | 95.90 |
| SVHN | single network (SVHN) | 92.94 | – | – |
| | single network (CIFAR) | – | 79.69 | – |
| ↓ | LF ($\lambda_e = 2.0 \times 10^{-3}$) | 83.18 | 48.99 | 66.09 |
| CIFAR-10 | LwF ($T = 1$) | 91.76 | 70.57 | 81.17 |
| | LwF ($T = 2$) | 90.19 | 71.94 | 81.07 |
| | PackNet | 92.84 | 76.78 | 84.81 |
| | HC-Net | 90.05 | 76.62 | 83.34 |
| | HC-Net (without H-Net) | 92.21 | 76.70 | 84.46 |

| Datasets | Methods | MNIST(%) | SVHN(%) | CIFAR(%) | Avg(%) |
|---|---|---|---|---|---|
| MNIST | single network (MNIST) | 99.44 | – | – | – |
| ↓ | single network (SVHN) | – | 92.94 | – | – |
| SVHN | single network (CIFAR) | – | – | 79.69 | – |
| ↓ | LF ($\lambda_e = 1.0 \times 10^{-3}$) | 89.57 | 72.1 | 45.43 | 69.03 |
| CIFAR-10 | LwF ($T = 1$) | 95.06 | 86.80 | 68.98 | 83.61 |
| | LwF ($T = 2$) | 86.09 | 85.07 | 69.63 | 80.26 |
| | PackNet | 99.38 | 91.93 | 66.34 | 85.88 |
| | HC-Net | 99.41 | 89.36 | 74.93 | 87.90 |
| | HC-Net (without H-Net) | 99.41 | 91.84 | 75.02 | 88.76 |

Table 3: Mean classification results on the basic datasets (5 runs).

baseline methods. Nonetheless, our full model, HC-Net also outperforms PackNet, especially on the second task by around 0.7%.

### 3.1.2 SVHN → CIFAR-10

In this experiment, we want to see the performance of the proposed method when the statistics of the two tasks are quite different. The datasets SVHN and CIFAR-10 have a clear distinction in image styles and colors. For LF whose classifier must be shared between the old and new tasks, this inevitably leads to a significant decrease in performance. LwF and PackNet show similar trends as that of 'MNIST to SVHN'. Higher temperature ($T$) induces better performance in the new task but a bit of degradation in average performance still occurs. PackNet experiences almost no performance degradation. The pruning method of PackNet acts like a generalization method and helps the model to maintain its capacity. The average performance of C-Net is slightly lower than that of PackNet by 0.35%. However, note that PackNet requires additional finetuning after the pruning process and takes additional time for training. On the other hand, C-Net requires no additional pruning or finetuning precedure and yet retains the performance.

### 3.1.3 MNIST → SVHN → CIFAR-10

Training three or more tasks is unmanageable for the models using regularization methods. LF scored an average accuracy of no more than 69.03%, which is much more lower than any other methods. No matter how we change the temperature ($T$), LwF suffers from a drop in average performance, especially in the third task. Also, even though the proportion allocated to each task is equivalent between the PackNet and our methods (HC-Net and C-Net), our method highly outperforms the PackNet in the third task. This implies that HC-Net is more suitable for multi-task sequential learning situation.

### 3.2 CONTINUAL LEARNING USING REALISTIC DATASETS

ImageNet contains images more realistic than other datasets used in this paper. Higher resolution and complex backgrounds make the classification even harder. To save the training time, two subsets of the ImageNet dataset each having 50 randomly chosen classes have been used for evaluation. They

| datasets | Methods | Old(%) | New(%) | Avg. (%) |
|---|---|---|---|---|
| ImageNet-A ↓ ImageNet-B | single network (ImageNet-A) | 83.28 | – | – |
| | single network (ImageNet-B) | – | 85.28 | – |
| | LwF ($T = 1$) | 82.2 | 86.72 | 84.46 |
| | LwF ($T = 2$) | 80.92 | 86.96 | 83.94 |
| | PackNet | 82.16 | 88.72 | 85.44 |
| | HC-Net (without H-Net) | 83.30 | 88.66 | 85.98 |

Table 4: Mean classification results for the realistic dataset (2 runs)

| Initialization | Old(%) | New(%) | Avg.(%) |
|---|---|---|---|
| Random | 99.42 | 91.67 | 95.55 |
| Pretrained parameters | 99.43 | 92.37 | 95.90 |

Table 5: Performances of the old (MNIST) and new (SVHN) tasks with different initialization methods: 1) initialization with only random Gaussian noise and 2) the pretrained parameter added by random Gaussian noise

are referred to as ImageNet-A and ImageNet-B respectively in this paper. To show the adaptability of our method on structures having shortcut connections, ResNet-50 is used for the experiment. We use the same experimental setting as in (He et al., 2016) with a batch size of 128. We compare our C-Net with LwF and PackNet.

Table 4 shows the results on the ImageNet dataset. The 'Old' accuracy of all methods slightly dropped from those of single networks. Like in the 'SVHN to CIFAR-10' experiment, PackNet shows a better result than the LwF regardless of the value of $T$. The result of C-Net is almost identical to PackNet with a slight increase in the average accuracy.

PackNet actually utilizes more parameters for each task than C-Net does. Since the masks force weights with no influence to be zero and utilize the well trained remaining weights, PackNet can make good use of the entire network. However in C-Net, just adding filters gradually without any other post-processing performs well enough compared to PackNet. When the model of an initially designed size is fully occupied by several tasks, C-Net can just add more filters and train them along with the trained filters while this is not the case of PackNet. Furthermore, PackNet requires additional memories to store the masks for all filters in it. C-Net needs only one integer per layer for each task. For these reasons, C-Net is far more efficient than the PackNet with no loss in performance.

## 3.3 ABLATION STUDY ON THE EFFICIENCY OF C-NET

As mentioned in Section 2.1, our C-Net appends filters whenever a new task is to be trained. In this process, parameters learned from previous tasks are utilized altogether and the parameters which are to be trained newly are initialized using the existing ones. We conduct experiments to show the effect of these methods. Experiments are carried out in 'MNIST to SVHN' case as a representative.

### 3.3.1 INITIALIZATION WITH PRETRAINED PARAMETERS

To analyse the effectiveness of initialization using pretrained old task parameters with random Gaussian noise, we compare the accuracy between a model initializing the new parameters with just a random Gaussian distribution and a model initializing the new task parameters using the pretrained old task parameters with additional random Gaussian noise. In Table 5, the model initialized by our method obviously shows higher accuracy on the SVHN task. This result implies that a good initialization prevents the model from going through a local minima.

### 3.3.2 PARAMETER SHARING

We verify the effect of the parameter sharing between the old task and the new task. The baseline method is a model where the old parameters are filled with random Gaussian initialization and no further training is done. The new parameters in the model have to learn knowledges from the new task without the aid of old parameters since they are fixed from the beginning. On the other hand, the old parameters in our method is trained by the old task and the new parameters can make use of these learnt knowledges to learn the new task. To solely observe the effect of the parameter sharing, both

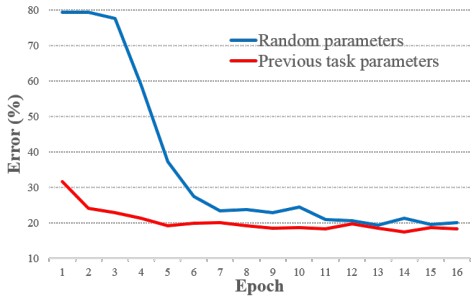

| #of increasing each index | Ours(%) | Random(%) |
|---|---|---|
| 2 | 80.60 | 78.10 |
| 4 | 85.31 | 83.85 |
| 6 | 86.98 | 86.01 |

Figure 3: Error rate comparison between sharing the parameters which learned previous task and random parameters.

Table 6: The performances on the new task while increasing the filter index of the model using parameter sharing and the model with fixed random parameters.

methods do not use our parameter initialization method mentioned in the previous section. The task indices $I^1$ and $I^2$ are {16,16,32} and {18,18,34} respectively. The model using parameter sharing converges far more faster than the model with no parameter sharing which is shown in Figure 3. Also the final error rate of our method is lower than the other. Increasing the number of indices elevation enhances our method by 6.38% and the other model by 7.91% shown in Table 6. This implies that the parameter sharing improves the model and allow it to be compact with fewer numbers of filters.

## 3.4 H-NET

The results of H-Net are reported in Table 2. In 'MNIST to SVHN', all methods approximates the results to perfection. Despite of the similar appearance of these two datasets, VAE has been able to generate distinct outputs. If a sample in the MNIST dataset is given, the result from the VAE trained for SVHN resembles those of the SVHN dataset. In other words, auto-encoders of each task tends to memorize only what they used for training. PCA is completely able to discriminate the origin of the images from MNIST and SVHN. Particularly in this experiment, even when using a single eigen-vector with the least eigen-value showed a result of 100% and 99.7%. According to the dataset, PCA can highly reduce the computational cost as well as the memory resource.

In 'SVHN to CIFAR-10', GAN showed the best discriminating performance. VAE shows a poor result on SVHN while the performance on CIFAR-10 remains more than 97%. This means that the auto-encoder trained for CIFAR-10 reconstructs SVHN images well, leading to low reconstruction losses in both of the auto-encoders. With PCA, both datasets are able to separate from each other with chances more than 92%. The experiments using all three datasets showed almost the same results as the experiments above. GAN definitely shows the best results among all methods introduced.

### 3.4.1 LIMITATION

Generative models usually have difficulties in generating realistic images with high resolution and there are few researches experimented on ImageNet data. This incompetence makes the H-Net hard to be applied to the ImageNet data. Also, as the subgroups share the same statistics, PCA hardly catch the characteristic of each set. Experiments have shown poor results on ImageNet and we leave this for a future work.

## 4 CONCLUSION

In this paper, we have proposed a novel framework which is able to divide its capacity into several parts and utilize them according to the given input. Composed of two networks, H-Net is responsible for memorizing "where the data came from" and complicated knowledges such as "what the data is" are engraved to the C-Net. To the best of our knowledge, our work, H-Net, is the first attempt to estimate the origin of data which used to be assumed as given in the previous works. As well as overcoming the catastrophic forgetting, HC-Net allows extra class labels when training a new dataset and also has expandability to the network architecture.

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

## APPENDIX A   H-NET

### A.1   VARIATIONAL AUTO-ENCODER (VAE)

This approach is based on the variational auto-encoder. VAE estimates the data distribution by approximating it to a Gaussian distribution. We trained auto-encoder for each task separately with the following reconstruction loss:

$$\mathcal{L}_r(AE_{\mathcal{J}}) = \|x - AE_{\mathcal{J}}(x)\|^2 \tag{3}$$

where $AE_{\mathcal{J}}(x)$ is the reconstructed data from the auto-encoder for the task $\mathcal{J}$. As $AE_{\mathcal{J}}$ is trained only with samples of the task $\mathcal{J}$, samples from other tasks will produce large reconstruction loss by $AE_{\mathcal{J}}$ since they will not follow the trained distribution. In the testing phase, an unknown input data is given to all auto-encoders. Among the results from the trained auto-encoders, the task $\mathcal{J}$ is determined to the one that has the smallest reconstruction error:

$$\mathcal{J} = \arg\min_i \|x - AE_i(x)\|^2. \tag{4}$$

### A.2   PRINCIPAL COMPONENT ANALYSIS (PCA)

Another approach to separate a dataset from another is using PCA. Typically, PCA conducts dimensionality reduction over a given dataset and the projected features that have the biggest eigen-values are used to represent each sample. The bigger the eigen-value is, the more the corresponding eigen-vector explains the variance of the dataset and the PCA-based reconstruction error similar to (4) can be used for the determination of the appropriate task.

In this paper, instead of using principal components of $top\text{-}k$ eigen-values, we utilized those of the $bottom\text{-}k$ eigen-values. A small eigen-value indicates low variance over the projected coordinate. Therefore, a sample from other tasks is likely to be far from the origin in this projection while a sample from the original dataset is near the origin. If the matrix of the eigen-vectors corresponding to the top $L$ smallest eigen-values is denoted as $\mathbf{W}_L^i$ which is extracted from the $i\text{-}th$ dataset, a sample $x$ from an unknown origin can be projected to produce the projection vector $\mathbf{T}_L^i = x^T \mathbf{W}_L^i$ and the task can be found as the one that minimize the $L_1$ or $L_2$ norm of the vector as

$$\mathcal{J} = \arg\min_i \|\mathbf{T}_L^i\|_p \tag{5}$$

where $p = 1$ or 2. In all our experiments, the number of eigen-vectors $\mathbf{L}$ is set empirically to 1,000.

## APPENDIX B   TRAINING DETAILS

### B.1   TRAINING DETAILS OF C-NET

We use the same learning parameters and network architectures suggested in (Jung et al., 2018) used for 'Tiny image classification'. It is composed of three convolutional layers using $5 \times 5$ kernels with the size of 32, 32 and 64 channels respectively, three max pooling layers, one fully connected layer with 200 nodes and the last softmax classifier layer producing 10 outputs. ReLU is used as the activation function in all the experiments. Also an SGD optimizer with a mini-batch size of 100 has been used for model optimization. When training the old task, the weight decay and the momentum were set to 0.004 and 0.9 respectively. The learning rate starts from 0.01 and the decay of the learning rate with a factor 0.1 is done at the time of 20,000 iterations. After 40,000 iterations, the training is terminated in all the experiments. When training a new task, the training of several approaches including Modified LwF, EWC and LF starts with a learning rate of 0.0001 and is terminated at 10,000 iterations without learning rate decay as in (Jung et al., 2018). Unlike above methods, LwF, PackNet and HC-Net have the same settings for the old and new task because they need more iterations as the new parameters are to be trained from the scratch. Note that PackNet needs additional pruning and finetuning step. In the new task training of LwF, we start with a learning rate of 0.0002 and 0.001 which are $0.1 \sim 0.02$ times less than that of old task for the 'MNIST to SVHN' and 'SVHN to CIFAR-10' respectively, as recommended in (Li & Hoiem, 2017). The channel split ratio which determines $I^{\mathcal{J}}$ follows the setting of PackNet.

### B.2 TRAINING DETAILS OF H-NET

### B.2.1 GAN

We use the deep convolutional generative adversarial networks (DCGAN) (Radford et al., 2015). In all experiments of the basic image classification dataset, we use the size of latent vector to 200. The generator use the four transposed convolution layers which are using 512,256,128 and 3 channels with $4\times4$ kernels, respectively and we use ReLU activation for all layers except for the output, which uses Tanh. Transposed convolution layers use 2 stride except the first layer. we use LeakyReLU activation in the discriminator for all layers. For the discriminator, the last convolution layer is flattened and then fed into a sigmoid output to discriminate. The architecture of discriminator is symmetrical to it of the generator. We use batchnorm in both generator and the discriminator.

We use the Adam optimizer with learning rate of 0.0002 using the batch size of 64. The momentum term $\beta1$ set to 0.5 and we train the generator until 200 epochs. The generated samples of basic image classification dataset are shown in Figure 4(a), 5(a) and 6(a).

For the binary classifier, we use the same network and experimental settings in (Jung et al., 2018) used for 'Tiny image classification'.

### B.2.2 VAE

We use the simplified VAE. we use two fully connected layer with a hidden layer with 400 units and 20 units for mean and variance of the latent state distributions. The decoder have an equal number of hidden units.

We train the VAE with ReLU activation, sigmoid activation and Adam optimizer with a learning rate of 0.0001 using batch size of 128. We train the VAE until 100 epochs.

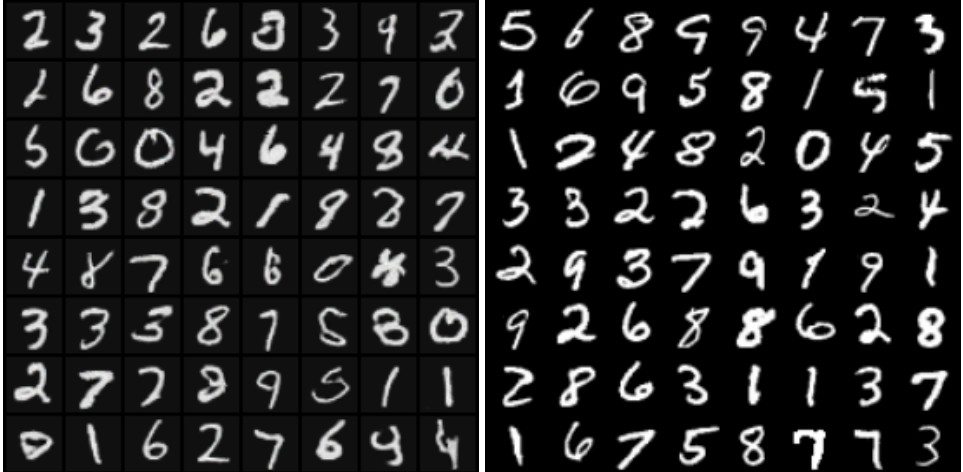

(a) Generated samples with the generator of H-Net

(b) Real samples

Figure 4: MNIST dataset

## APPENDIX C   RELATED WORK

Dual memory network system has been suggested to overcome the low performance of the regularization methods and the inefficiency of the dynamic methods. Most of these researches try to mimic the brain system of mammals whose recognition process is believed to be done by the complementary contribution of the hippocampus and the cortex. In several researches, the hippocampus has been realized by networks replaying the previous dataset or determining whether the input is from the old dataset (Gepperth & Karaoguz, 2016; Kemker & Kanan, 2017). Shin et al. (2017) proposed a

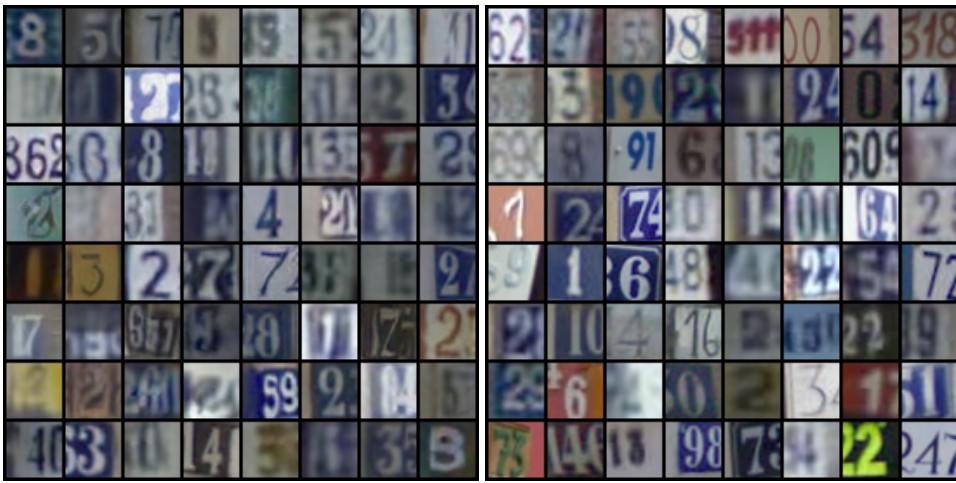

(a) Generated samples with the generator of H-Net

(b) Real samples

Figure 5: SVHN dataset

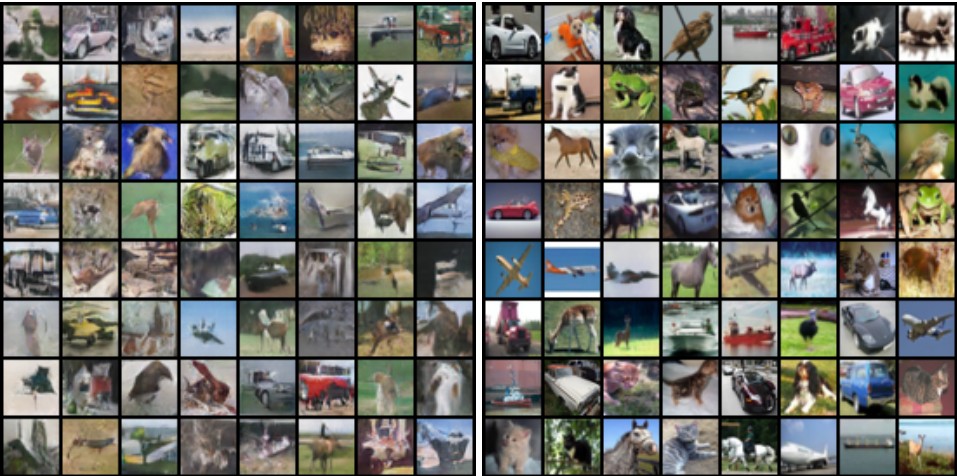

(a) Generated samples with the generator of H-Net

(b) Real samples

Figure 6: CIFAR-10 dataset

method of replaying previously encoded experiences using GAN and training the new task without the presence of the old dataset.

In the data mining field, multi-task learning can be seen as learning from data streams. Therefore, our method can be considered to be one of researches on learning methods from data streams (Gama & Gaber, 2007; Gama, 2010).

