# OpenReview forum: "HC-Net: Memory-based Incremental Dual-Network System for Continual learning"
_ICLR.cc/2019/Conference_

### Official Review · AnonReviewer3 · 2018-10-23
**Not enough novelty and experiments are not very informative.**

**Rating:** 4
**Confidence:** 5

**Review:**

The key idea of this paper is to expand the network for training on new tasks which is termed as C-Net, and train an additional generative model which is used for predicting task id (which is called H-Net), and use the task id for selecting weights from the C-Net.

Pros:
1. It is relatively easy to understand the paper.
2. The originality of this paper lies in the usage of generative model to predict task id (H-Net). To my knowledge has not been proposed before.
3. In contrast to previous works in multi-task learning, which assumes task id is available both during training and inference, this work tries to remove the need of task id during inference, which makes it closer to the general definition of continual learning.

Cons:
1. Expanding the network for new tasks is not a novel contribution of this paper, it has already been proposed in previous works on multi-task learning. Doing expansion on all of the layers does not qualify for a major contribution in my opinion.
2. The experimental comparison is not very fair in my opinion,
     a. Comparing accuracy of C-Net to other methods is not very useful. Because this methods expands the network for every new task, while other methods (EWC, LwF) has limited to no expansion in the network. Given that the single network result is far from state of the art (table 3), I suppose model size could contribute to the accuracy boost.
     b. It is not explicitly stated in the paper whether the output neurons are shared between tasks or an individual set of output neurons are used for different tasks, but from the rest of the paper I suppose disjoint neurons are used. Then the comparison between EWC and this work is not fair because EWC shares the output neurons among tasks.
This is not to blame this paper for not making fair comparison, since given different assumptions between methods (availability of task id, shared output neurons etc.), it is usually difficult to fairly compare between continual learning methods.  This problem is raised in another submission https://openreview.net/forum?id=ByGVui0ctm. The point here is that the accuracy of C-Net is not a good measure of how good this method is.
3. I disagree with the point that MNIST and SVHN are similar, they have very different distributions and are very easy to tell apart with a model. One concern is that the generative H-Net may fail to work once the distributions of the tasks overlap to some extent. e.g. cifar10 vs cifar100.

As a conclusion, the key contribution of this work is using generative model to determine task id which removes the need for task id during inference. It is relatively insufficient for publication on ICLR.

---

> ### Author Response · Authors · 2018-11-26
> **Response to reviewer 3**
>
> Thank you for the comments and feedbacks.
>
> 1. The major comparison in this paper was between the Packnet and ours. We wanted to emphasize that our model can avoid catastrophic forgetting and 'also' can expand if needed. We agree that we should have been careful in writing the contribution.
>
> 2. (a) Maybe there was a misunderstanding in this part. The overall architectures are the same in all experimental settings(EWC, LwF, Packnet and ours). So we are not giving more capacity to our network but efficiently distributing the capacity to each task just like the Packnet. In other words, we do not expand the c-net capacity compared to the baseline network.
> As we are not using the full Imagenet dataset, we believe that the result in Table 3 may be different with the full dataset result. We assumed that Resnet-50 is enough to show the properties of our method.
>
> (b) We agree with your opinion about EWC and we are willing to remove it from the paper. Except that, we want the readers understand that our method has the same or better performance with the Packnet. Even so, our method does not need masks in all filters and has flexibility.
>
> 3. We agree with your opinion and that is what we noted in the limitation section.

---

> > ### Comment · AnonReviewer3 · 2018-12-09
> > **Thanks for clarification**
> >
> > I'm keeping the evaluation that the contribution is not sufficient for ICLR.

---

### Official Review · AnonReviewer2 · 2018-11-01
**missing references and comparisons, unclear evaluation**

**Rating:** 4
**Confidence:** 4

**Review:**

Summary
This paper proposes an extension of Progressive Networks [Rusu et al. NIPS 2016] (unfortunately, not cited) where the task id is not given at test time. This is inferred by a battery of classifiers trained on data produced by generative models trained on task specific data.
The authors argue strong connections to similar mechanisms in the human brain and demonstrate this method on a stream of 2 or 3 vision tasks. However, the interpretation of these results is dubious.

Novelty: given prior work on Progressive Nets and other methods using generative models for continual learning, novelty is limited.

Relevance: the motivation and aim of this work is certainly relevant for ICLR.

Clarity: the paper is overall clear, although it needs a bit of rewriting to improve fluency (see for instance sec. 3.4.1).

References: the authors should definitely cite Progressive Networks and their extension "Progress and Compress" (Schwarz ICML 2018), as their approach is an extension of the former with the only difference that the task id is inferred at test time by using a battery of binary classifiers.

Empirical validation: The empirical validation is limited because of:
a) lack of comparison to Progressive Nets,
b) lack of simple baselines (e.g., how about replacing H-Net with an inference process like task_id = argmin_i=1..T loss(C-Net, task = i) ),
c) unclear interpretation of the provided results (how can the accuracy on MNIST be 100%? are the authors reporting training accuracy?)
d) very limited number of tasks considered (up to 3)

General comments
Major drawbacks of the proposed approach are: 1) training on new tasks can never improve performance on past tasks (unlike other methods like GEM (Lopez-Paz et al. NIPS 2017), 2) the number of parameters grow linearly with the number of tasks (an issue addressed by the Progress and Compress paper above), and 3) the overall approach is not efficient as it requires lots of data from each task in order to train the generative models.
Finally, I think all the connections and inspiration from how the human brain works should be toned down.  Statements like "the C-Net corresponds to the human cortex..." should be at the very least rephrased appropriately.

---

> ### Author Response · Authors · 2018-11-26
> **Response to reviewer 2**
>
> Thank you for the comments and feedbacks.
>
>  We agree that the citation of Progressive networks should be included in this paper. However, What we meant in this paper was that  our method can allocate each task parameters to each part of the model and that is what the Packnet does. We simply wanted to show that given the same number of parameters, additional pruning process and filter mask of Packnet are not necessary. Also, progressive network includes additional adaptor modules in each layer and is difficult to say which setting would contain the exactly equal capacity of our model.  These are the reasons why we presented a comparison with the Packnet.
>
> (a) We agree that the citation of Progressive networks should be included in this paper. However, What we meant in this paper was that  our method can allocate each task parameters to each part of the model and that is what the Packnet does. We simply wanted to show that given the same number of parameters, additional pruning process and filter mask of Packnet are not necessary. Also, progressive network includes additional adaptor modules in each layer and is difficult to say which setting would contain the exactly equal capacity of our model.  These are the reasons why we presented a comparison with the Packnet.
> (b) We agree that your suggestion is reasonable.
> (c) The results are evaluated in the validation data of each dataset. Separating Mnist from SVHN is obviously an easy task for the Hnet and we wrote the details in the 'Hnet' section.
>
> (d) With the lack of time and resources, we evaluated our method on three tasks. We can try more tasks in the future but we think that this won't change the aspects of our experiments.
>
> (1) Instead of improving the old tasks performance we have chosen to 'not degrading' the performance. This point can samely applied to the Packnet.
> (2) Given the same parameter, we can allocate the model's capacity to each task and 'also' can expand the network if needed, which is impossible for the Packnet.
> (3) The Hnet is trained with each dataset in return for the ability to know where the data comes from. We think that whether this approach is efficient or not is a subjective problem. In the previous works such as Progressive networks, LwF and Packnet, they just left this problem as a future work and we just suggested a baseline method for it.
> (4) We admit that the expression about the human brain may be too exagerated.

---

> > ### Comment · AnonReviewer2 · 2018-11-27
> > **thank you**
> >
> > Thank you for the clarifications. I still think the paper will need a major revision to address all the issues we discussed.

---

### Official Review · AnonReviewer1 · 2018-11-10
**Not good enough**

**Rating:** 4
**Confidence:** 3

**Review:**

The work proposes a structure that mimics progressive nets. Maybe the main difference from progressive nets is that backwards connection from the new features to the old features in layer 2 are not 0 out. This could cause interference, however is solved by using the task ID to not evaluate those new features when going back to a previous task. I think this is a technical detail, that does not provide any explicit advantage or disadvantage over progressive nets.

Employing GANs/VAE to predict task id also can be seen as not an ideal choice. In particular the GAN network will suffer from catastrophic forgetting, which is solved (if I understood correctly) by training the GAN with data from all tasks. Which makes one wonder, if we can affort to access data from all tasks to learn the GAN then why not the classification model too !?

I think an alternative might be something like the Forget Me Not Process published and used in the original work with EWC.

Unfortunately due to presence of these previous works, lack of more thorough comparison with other existing approaches, the work should not be accepted to ICLR.

---

> ### Author Response · Authors · 2018-11-26
> **Response to reviewer 1**
>
> Thank you for the comments and feedbacks.
>
> We think there was a misunderstanding. Gradients to the old features are 0 out to make sure they do not ruin the trained parameters.
> In the progressive network, each feature is summarized by an additional adapter and passed to the next task network. Therefore, it is not a typical form of CNN. Meanwhile, our HCnet takes the form of a regular CNN and just expands the number of filters so that the new information is learned by those additional filters. That is why we compared our method to the Packnet. Packnet also divides its capacity into several pieces but has no ability to add more tasks once the network is trained and pruned. What we wanted to present in this paper was (1) we can retain the properties of the Packnet (2) without using any pruning process which makes the packnet unexpandable, (3) in a typical form of CNN so that it can be easily used in various applications.
>
> In the case of the GAN/VAE, We do not use the whole datasets for the generative model. If there are three tasks, the H-Net also has three generative models. Each generative model is trained only using each task data. Then, the binary classifier use the three generators for training.

---

### Meta-Review · Area_Chair1 · 2018-12-15

**Confidence:** 5
**Recommendation:** Reject

**Metareview:**

This work is effectively an extension of progressive nets, where the task ID is not given at test time. There were several concerns about novelty of this work and the evaluation being insufficient. There was a reasonable back and forth between the reviewers and authors, and the reviewers are all aligned with the idea that this work would need a substantial rewrite in order to be accepted at ICLR.